# Metformin Modifies the Gut Microbiota of Mice Infected with *Helicobacter pylori*

**DOI:** 10.3390/ph14040329

**Published:** 2021-04-03

**Authors:** Marine Jauvain, Sarah Courtois, Philippe Lehours, Emilie Bessède

**Affiliations:** 1INSERM, University of Bordeaux, UMR 1053 Bordeaux Research in Translational Oncology, BaRITOn, F-33000 Bordeaux, France; marine.jauvain@u-bordeaux.fr (M.J.); sarah.courtois@u-bordeaux.fr (S.C.); philippe.lehours@u-bordeaux.fr (P.L.); 2French National Reference Center for Campylobacters & Helicobacters in Bordeaux (CNRCH), University Hospital of Bordeaux, 33000 Bordeaux, France

**Keywords:** *Helicobacter pylori*, metformin, microbiota, PICRUSt

## Abstract

Metformin is widely prescribed to treat type 2 diabetes. Diabetes patients treated with metformin have a decreased risk of cancers, including gastric cancer. Among the factors influencing digestive carcinogenesis, gut microbiota interactions have been intensively studied. Metformin exhibits direct antimicrobial activity toward *Helicobacter*
*pylori*, which plays a crucial role in gastric carcinogenesis. Mice were infected with *H. pylori* and treated for 12 days with either metformin or phosphate-buffered saline (PBS) as a control. At the end of the treatment period, the mice were euthanized and cecal and intestinal contents and stool were collected. The gut microbiota of the three different digestive sites (stool, cecal, and intestinal contents) were characterized through 16S RNA gene sequencing. In mice infected with *H. pylori*, metformin significantly decreased alpha diversity indices and led to significant variation in the relative abundance of some bacterial taxa including *Clostridium* and *Lactobacillus*, which were directly inhibited by metformin in vitro. PICRUSt analysis suggested that metformin modifies functional pathway expression, including a decrease in nitrate reducing bacteria in the intestine. Metformin significantly changed the composition and predicted function of the gut microbiota of mice infected with *H. pylori*; these modifications could be implicated in digestive cancer prevention.

## 1. Introduction

Metformin, also known as 1,1-dimethylbiguanide, is the most widely prescribed glucose metabolism regulator for the treatment of type 2 diabetes mellitus globally [1]. The pharmaceutical effect of metformin is partially determined by AMP-activated protein kinase (AMPK) activation [2]. In response, the digestive system modifies glucose absorption and enhances anaerobic glucose metabolism [3]. In animals, metformin is accumulated at very high concentrations in the wall of the intestine [4]. For several years metformin has also been studied intensively for its antitumor properties in different types of cancer [5].

In 2005, a Scottish study hypothesized that metformin treatment may reduce cancer risk in diabetic patients [6]. A meta-analysis of seven cohort studies of gastric cancer, which has the third greatest mortality among cancers worldwide [7], showed that gastric cancer risk decreased in diabetic patients treated with metformin [8]. *Helicobacter pylori* plays a crucial role in gastric carcinogenesis by promoting inflammation and degradation of the gastric epithelium [9]. This Gram-negative bacterium colonizes the stomach mucosa in more than 90% of all gastric cancer patients [10]. *H. pylori* infection is among the most prevalent infections worldwide [11]; however, only 1% of infected patients develop gastric adenocarcinoma [12]. Factors influencing gastric cancer occurrence in infected patients include genetic host factors and environmental factors including the host microbiota [13].

The gut microbiota have many functions, including intestinal homeostasis regulation, immune response, and metabolism energy regulation [14]. Modification of the gut microbiota can be associated with a number of diseases, including cancer [15]. In colorectal cancer, bacteria composing the digestive microbiota help to form the micro-tumoral-environment and may influence cancer development [16]. *H. pylori* colonization influences the gastric microbiota; its infection leads to a significant decrease in gastric bacterial diversity [17]. Changes in the gastric microbiota with the evolution of gastric mucosa were shown including mucosal microbiome dysbiosis in gastric carcinogenesis [18]. Metformin has been shown to inhibit *H. pylori* growth directly in vivo and in vitro. Mice infected with *H. pylori* and treated with metformin show a decrease in gastric *H. pylori* colonization [19]. Therefore, in the present study, we hypothesized that metformin may affect other bacteria, especially gut bacteria.

In this context, the aim of this study was to describe the effects of metformin on the gut microbiota in *H. pylori*-infected mice. We studied metformin-induced changes in the composition and metabolic functions of the gut microbiota in three different digestive sites (stool, cecal, and intestinal contents), in mice infected with *H. pylori* that were treated with either metformin or phosphate-buffered saline (PBS) as a control. 

## 2. Results

### 2.1. Alpha Diversity Indices Are Reduced in the Gut Microbiota of Mice Infected with H. pylori and Treated with Metformin

The Chao1, Shannon, and phylogenetic diversity (PD) whole tree alpha diversity indices were used to characterize the richness and diversity of the bacterial community within each sample. At the end of the treatment period, alpha diversity indices were compared between the metformin and control groups for three different sample types: stool, cecal content, and intestinal content.

In stool, metformin treatment induced a significant reduction in alpha diversity indices (*p* ≤ 0.002; Figure 1). These significant decreases were also observed in the other anatomical digestive sites studied (cecum and intestine). Thus, metformin treatment led to reductions in all alpha diversity indices studied in a homogenous manner throughout the mouse digestive system. 

### 2.2. Beta Diversity Analysis Shows Changes in the Gut Microbiota of Mice Infected with H. pylori and Treated with Metformin

Beta diversity analysis of the three types of digestive samples clearly showed that the metformin and control treatment groups clustered separately at each digestive site (Figure 2 and Appendix A). Weighted UniFrac distances showed that the total diversity values of the two principal coordinates were 74.91%, 75.05%, and 85.53% for stool, cecal, and intestinal content, respectively. 

A comparison of bacterial profiles based on weighted and unweighted UniFrac and Bray–Curtis distances showed significant differences between the metformin and control groups (Figure 2 and Appendix A). Adonis statistical tests showed that these observed differences were statistically significant (*p* = 0.001 for weighted UniFrac, unweighted UniFrac, and Bray–Curtis distances). Thus, metformin induced significant modification of the gut microbiota composition in *H. pylori*-infected mice.

### 2.3. Metformin Treatment Changes Taxonomic Repartition in the Gut Microbiota of Mice Infected with H. pylori

The microbial compositions of the three digestive sites at the phylum, class, order, and family levels are shown in Table 1. Only taxa with a relative abundance of ≥0.1% were computed. Bacterial taxa with the most significantly different microbial abundance between treatment groups (*p* < 1.10^−4^) are highlighted. 

At the phylum level, fecal microbiota was dominated by *Firmicutes* in both groups, followed by *Actinobacteria* in the metformin group (8.52% control vs. 28.76% metformin, *p* = 5.6 × 10^–4^) and *Bacteroidetes* in the control group (17.94% control vs. 6.50% metformin, *p* = 6.8 × 10^–5^). The × same trend occurred in cecal microbiota composition. In the intestinal microbiota, the control group was dominated by *Firmicutes* (54.14% control vs. 22.63% metformin, *p* = 4.2 × 10^–5^), whereas the metformin group was dominated by *Actinobacteria* (37.08% control vs. 70.97% metformin, *p* = 1.5 × 10^–5^). Metformin treatment decreased abundance of *Firmicutes* and *Bacteroidetes* for the benefit of increased *Actinobacteria,* in the three digestive sites.

Notably, metformin treatment significantly decreased the abundance of *Clostridiales* in the cecum and intestine (90.33% control vs. 72.38% metformin, *p* < 1 × 10^–6^, 25.2% control vs. 2.89% metformin, *p* = 9 × 10^–6^ in cecal and intestinal samples, respectively). However, in the three digestive sites, metformin treatment increased *Bifidobacteriales* abundance (7.28% control vs. 28.14% metformin, *p* = 2.83 × 10^–4^, 1.39% control vs. 18.15% metformin, *p* = 2 × 10^–6^, 20.79% control vs. 67.44% metformin, *p* < 1 × 10^–6^, in stool, cecal, and intestinal samples, respectively).

Taxonomic repartition at the family level is shown as bar plots in Figure 3a. Interestingly, family *Bifidobacteriacea* was significantly more abundant in the microbiota of the metformin group than in that of the control in all three digestive sites (7.3% control vs. 28.1% metformin, *p* = 2.8 × 10^–4^, 1.4% control vs. 18.2% metformin, *p* = 2 × 10^–6^; 20.8% control vs. 67.4% metformin, *p* < 1 × 10^–6^ in fecal, cecal, and intestinal samples, respectively). Metformin treatment significantly decreased *Lachnospiraceae* abundance in cecal and intestinal contents, but this effect was not observed in fecal samples (34.13% control vs. 34.93% metformin, non-significant, 57.86% control vs. 43.3% metformin, *p* = 3.5 × 10^–4^; 19.8% control vs. 2.11% metformin, *p* = 9 × 10^–6^ in fecal, cecal, and intestinal samples, respectively; Table 1, Figure 3a).

Liner discriminant analysis (LDA) effect size analysis (LEfSe) was conducted to identify differentially abundant taxa in the three sample types. LDA scores were determined, and the specific taxa associated with metformin treatment were identified (Figure 3b). Among the fecal samples, 16 bacterial taxa were identified, including 12 genera that were differentially abundant between treatment groups (Figure 3b). Compared to the metformin group, 13 taxa were more abundant in the fecal microbiota of control mice.

In the metformin group, a higher abundance of *Akkermansia*, *Anaerotruncus*, and *Bifidobacterium* genera were observed in the fecal microbiota and also in the cecal content. Among them, *Bifidobacterium* was also more abundant in intestinal microbiota (Figure 3b).

Within the genera Akkermansia and Bifidobacterium, the species Akkermansia muciniphila and Bifidobacterium pseudolongum were identified.

Appendix A shows the LEfSe analysis results for the different digestive sites. Only bacterial taxa with LDA scores > 2 in at least two of the three digestive sites were included; this criterion was met by 12 bacterial taxa in the control group and three bacterial genera in the metformin group (*Bifidobacterium*, *Anaerotruncus*, and *Akkermansia*). Operational taxonomic units (OTUs) number associated to these taxa were listed in Appendix A.

### 2.4. Metformin Directly Inhibits the Lactobacillus and Clostridium Gut Bacterial Strains In Vitro

To determine which bacterial strains are directly affected by metformin in the gut, we further examined those strains with significantly different relative abundance between the metformin and control groups and easily cultivable. Metformin treatment decreased the abundance of *Lactobacillus*, *Aerococcus*, and *Clostridiales* strains, and increased that of *Bifidobacterium* strains (Appendix A). 

We observed no significant growth differences in *Aerococcus sanguinicola*, *Bifidobacterium breve*, or *Bifidobacterium longum* at different metformin concentrations (*p* > 0.01, Figure 4). Significant decreases in growth were observed in *Lactobacillus harbinensis* and *Clostridium difficile* strains incubated with metformin concentrations of 20 and 50 mM (*p* < 0.01) and in *Clostridium perfringens* strains treated with higher metformin concentrations (50 and 100 mM, *p* < 0.01, Figure 4).

### 2.5. Potential Functional Pathways of Gut Microbiota Are Modified by Metformin Treatment in Mice Infected with H. pylori

Functional analysis of the gut microbiota was performed using the Phylogenetic Investigation of Communities by Reconstruction of Unobserved States (PICRUSt) software based on closed-reference selection of operational taxonomic units (OTUs). We examined 159 pathways based on the Kyoto Encyclopedia of Genes and Genomes (KEGG) reference database. Only KEGG pathways with a relative abundance > 0.001% were considered; these represented 124, 119, and 132 KEGG pathways in stool, cecal, and intestinal samples, respectively. Pathways with a significantly different abundance between the metformin and control groups were identified for all three different digestive sites (*p* < 0.05, after Bonferroni correction). These differentially enriched KEGG pathways are shown in Figure 5 and Appendix A.

Beta diversity analysis according to the predicted functional pathways for each group was performed. A comparison of the microbiota of the metformin and control groups based on metabolic function showed differential bacterial profiles on 2D PCoA plots (Figure 6a). Adonis statistical tests performed on these data showed significant differences among Bray–Curtis distances in all sample types (*p* = 0.001).

In stool samples, five predicted KEGG pathways were significantly more abundant in the metformin group, compared with eight KEGG pathways in the control group. The differential KEGG pathways between the metformin and control groups represented 10.5% and 26.9% of all pathways examined for stool and cecal content, respectively. The highest number of differential KEGG pathways was found in intestinal content (53.8%, Figure 6b). 

In intestinal samples, the most significantly enriched KEGG pathways in the metformin group were nicotinate and nicotinamide metabolism, peptidoglycan biosynthesis, secondary bile acid biosynthesis, and streptomycin biosynthesis (Figure 5). Carbohydrate metabolism was predicted to be higher in the microbiota of control group mice than in metformin-treated mice. Pathways implicated in carbohydrate metabolism (green text, Figure 5) including pyruvate, propanoate, ascorbate, butanoate, and glyoxylate metabolism were predicted to be overexpressed in the control group. 

Finally, nitrate and nitrite-reducing bacterial species were specifically studied in intestinal content by focusing on the KEGG gene expression of k02575, k00370, k00363, and k03385, which code for nitrate or nitrite-reductase enzymes. Intestinal bacteria in the microbiota of metformin-treated mice showed significantly decreased nitrate and nitrite-reductase functions (*p* < 0,05, Mann–Whitney *U* test) (Figure 6c).

PICRUSt analyses showed that metformin treatment led to significant changes in predicted metabolic functions by gut bacteria in infected mice, specifically in intestinal sites.

## 3. Discussion

In the present study, we examined changes in the gut microbiota at three different digestive sites, represented by stool, cecal, and intestinal samples, induced by oral metformin treatment of mice infected with *H. pylori*. We performed 16S rRNA gene sequencing and characterized the gut microbial profiles of mice infected with *H. pylori* and treated with metformin. 

Our results showed that metformin decreased richness and diversity of the microbiota of mice. High microbiota diversity and richness are usually considered to be markers of a healthy microbiota; however, decreases in the abundance of any bacterial taxa may lead to the relative emergence of metabolically beneficial microorganisms such as *Akkermansia muciniphila*, which is associated with metabolic improvement [20]. In this study, *Akkermansia muciniphila* was found to be more abundant in metformin-treated mice than in control mice, which is consistent with the results of a previous study of diabetic patients that suggested that these effects could contribute to the therapeutic effect of metformin in diabetes treatment [21]. 

Beta diversity analyses showed that microbial features depended on the treatment received. Other studies have reported significant changes in the microbiota of obese mice, healthy mice, and diabetic humans following metformin treatment [21,22,23,24]. 

The direct effect of metformin on bacterial growth was measured on six bacterial strains. Thanks to taxonomic composition analysis, we selected six bacterial species that were available in the laboratory, easily cultivable, and with relative abundance either positively or negatively influenced by metformin treatment. This experiment showed that metformin directly inhibited the growth of *Lactobacillus* and *Clostridium* gut bacteria. Metformin can also indirectly modify the microbiota by acting on host physiology; for example, metformin increases the bile acid pool within the intestine [25], which may affect stool consistency and the microbiome [26]. More recently, metformin treatment was revealed to enhance the release of glucose into the intraluminal space of the intestine in humans [27]; therefore, high glucose concentration in the intestinal lumen may impact bacterial development. Thus, metformin-induced microbiota changes are probably the result of both direct and indirect effects. 

The effects of metformin on human health have been intensively studied in recent years. Beyond its implication in diabetes treatment, metformin represents a promising anticancer drug in combination with conventional chemotherapies for different types of cancer [28]. Recent studies have also demonstrated the antiaging effects of metformin [29] and its direct antimicrobial effect against *H. pylori*, which has opened new avenues of research [19]. In cancer prevention, metformin has been shown to reduce cancer incidence in diabetic patients [6]. Microbiota composition and function are well recognized as influencing carcinogenesis through different mechanisms [30]. *H. pylori* is the best example of a specific bacterial pathogen that can trigger carcinogenesis by promoting inflammation and degradation of the gastric epithelium [9]. The bacterial microbiota may also influence intestinal barrier preservation, inflammation modulation, and the production of cancer-promoting metabolites [31]. In this context, we investigated the influence of metformin treatment on microbiota mechanisms potentially implicated in gut carcinogenesis.

Concerning specific bacterial taxa, despite decreases global richness, metformin treatment led to increases in *Bifidobacterium* abundance. *Bifidobacterium* species have demonstrated anti-colorectal cancer activity by producing metabolites that directly inhibit the growth of colon cancer cells in vitro [32]. *Bifidobacterium* species are often integrated into probiotic products for health treatments, including cancer prevention. It has been suggested that probiotics containing *Bifidobacterium* species can contribute to colorectal cancer prevention and improvement of safety and effectiveness of colorectal cancer therapy [33].

A recent study of diabetic and non-diabetic mice with induced colorectal cancer showed that metformin treatment in association with probiotics containing *Bifidobacterium* species actively prevented inflammatory and carcinogenic processes [34]. Furthermore a study of *H. pylori*-related gastric lesions showed a higher relative abundance of *Firmicute* in gastritis and gastric metaplasia patients [35]. Interestingly, our results showed a decreased relative abundance of *Firmicute* bacteria in *H. pylori*-infected mice in response to metformin treatment.

Functional features of the microbiota of mice in this study were examined using the PICRUSt software [36]. The resulting bacterial predicted profiles showed that the metformin and control groups had distinct metabolic functional signatures. The intestinal microbiota showed the highest expression among differential metabolic KEGG pathways between groups, indicating that metformin treatment leads to significant modification of the functional properties of the digestive microbiota, particularly in intestinal sites.

Specifically, metformin treatment decreased nitrate and nitrite reductase functions in intestinal bacteria. The nitrate-reducing bacterial pathway was analyzed because it has been suggested to participate in the increase of intragastric concentrations of nitrite and N-nitroso-compounds [37]. N-nitroso-compounds promote mutagenesis and protooncogene expression, and inhibit apoptosis; they can also contribute to gastric carcinogenesis [38,39]. Increased functional activity of nitrate reductase has been observed in the gastric microbiota of gastric cancer patients in comparison with chronic gastritis patients [40]. In the present study, KEGG pathways involved in carbohydrate metabolism were enriched in the control group, which comprised mice infected with *H. pylori* but not treated with metformin. These pathways are predictive of bacterial production of short-chain fatty acids [41], which have been linked to cell hyperproliferation in colorectal and esophageal cancer [42,43]. These pathways have also been found to be enriched in the gastric microbiota of gastric cancer patients [18]. Together, these findings demonstrate the potential contribution of bacteria-producing short-chain fatty acids to digestive tumorigenesis. These first results suggest that metformin, by modulating microbiota function, could be considers as a potentially interesting agent for digestive cancer prevention. Molecular mechanisms that sustain the anticancer effect of metformin through the regulation of glucose metabolism have been reported [44]. Thus, host physiology and the microbiota constitute different potential targets for metformin action in preventing cancer occurrence. In diabetic patients, metformin reduced the incidence of adenomas that could transform into colorectal cancer; therefore, metformin may be useful for the prevention of colorectal cancer in patients with type 2 diabetes [45]. In mice, metformin use in association with probiotics reinforces beneficial effect on colorectal cancer prevention [34].

The limitations of this study were the lack of information about gastric microbiota; more experiments should be performed to understand the metabolic modifications induced with metformin microbiota changes. We used female mice, which are less aggressive than male and easier to use in animal facilities. Consequently, results obtained are only valid in female and cannot completely be extrapolated to male as there are few differences in the composition of gut microbiota between genders and between female of different hormonal status [46,47]. However, a female from either the control or metformin group had the same age at the beginning and during all the length of the experiment; therefore, mice from the two groups were exposed to the same sexual hormonal modifications, allowing the groups comparison.

In conclusion, the results of this study show that metformin significantly alters the composition and predicted function of the gut microbiota of mice infected with *H. pylori*. These modifications could be implicated in gut cancer prevention.

## 4. Materials and Methods

### 4.1. Animal Protocol and Sample Collection

The animal protocol used in this study was previously described [19]. Five-week-old Specific Pathogen-Free C57Bl6J female mice were chosen for their better ability to live with partners. Mice were infected intragastrically on 3 consecutive days with 0.1 mL of a highly concentrated suspension of mouse-adapted *H. pylori* strains SS1 and B47 (Mc Farland 7 opacity standard) [48,49]. Three days after the last infection, the mice were divided randomly into two groups: an infected group treated with PBS as a control (*n* = 18, two mice died before the beginning of treatment) and an infected group treated with metformin (Sigma Aldrich, St. Louis, MO, USA) at 10 mg/mouse (*n* = 20). This dosage was determined using the method of dose conversion between human and animal studies [50]. With this method, 10 mg/mouse/day corresponds to 2.4 g of metformin/day for a human adult. The maximal dosage of metformin used to treat Type 2 diabetes patients is 3 g/day. Each group received a daily treatment (0.1 mL) for 12 days by gavage. During this period all mice had access to water ad libitum and a normal diet. Stool samples were collected before infection with *H. pylori* and after 12 days of treatment. Weight was controlled during the study. After the treatment, no differences were observed in mice weight between the two groups (data not shown). At the end of the treatment, mice were euthanized by cervical dislocation. Cecum and intestine were aseptically taken to collect cecal and intestinal content separately. Intestinal content corresponds to the entire content of mice intestine, no specific region in intestine was selected. Gastric samples were not available for use in this study.

All collected samples were immediately stored in a sterile tube at −80 °C. Mouse experiments were performed in level 2 animal facilities at Bordeaux University with the approval of the local Ethical Committee, and in conformity with the French Ministry of Agriculture (approval no. 4608).

Initially, alpha and beta diversity were analyzed using mouse stool samples collected from both groups prior to infection and treatment. This analysis confirmed that the groups were comparable and presented no differences in alpha or beta diversity (Appendix A). Same analysis was also performed on stools collected after infection with *H. pylori* and before any treatment with the same results (data not shown).

### 4.2. DNA Extraction and rRNA Gene Sequencing

DNA was extracted from samples using the QIAamp PowerFecal Pro DNA kit with a PowerLyzer (Qiagen, Hilden, Germany) according to the manufacturer’s protocol. Quantification, sequencing of the V3–V4 region of 16S rRNA, and assembly were performed by Genoscreen (Lille, France; further details provided in Supplementary material S1). The 16S rRNA sequencing datasets generated in this study can be found in the SRA database (http://www.ncbi.nlm.nih.gov/bioproject/701274, accessed on 3 April 2021).

### 4.3. Functional Metagenome Predictions

Phylogenetic Investigation of Communities by Reconstruction of Unobserved States (PICRUSt) 1.1.0 software was used to predict virtual metagenomes for each sample using the 16S rRNA gene sequencing results [36]. The Kyoto Encyclopedia of Genes and Genomes (KEGG) was used as a reference database. Based on the predicted metagenomes, the relative abundance of KEGG genes or KEGG pathways (ko) within each sample was determined.

### 4.4. Bioinformatics Analysis

Alpha and beta diversity were computed using the QIIME v1.9.1 software. The samples have been rarefied to 32,606 sequences for these analyses. Alpha diversity was calculated in terms of the Chao1, Shannon, and phylogenetic diversity (PD) whole-tree metrics. Beta diversity was calculated using weighted and unweighted UniFrac or Bray–Curtis distances. Ordination was performed using principal coordinate analysis (PCoA). The strength and statistical significance of beta diversity were computed using the Adonis method with QIIME.

Statistically significant differences in the relative abundance of taxa associated with the treatment groups were detected using linear discriminant analysis (LDA) effect size (LEfSe) [51]. Only taxa with LDA > 2 and *p* < 0.05 were considered significantly enriched.

Predicted functional genes were compared between groups and the results visualized using the STAMP v2.1.3 software [52]. Statistical differences in KEGG pathway frequencies were determined using White’s nonparametric *t*-test, followed by Bonferroni correction to adjust *p* values.

### 4.5. In Vitro Bacterial Growth Experiments

Strains of Aerococcus sanguinicola, Lactobacillus harbinensis, Bifidobacterium longum, Bifidobacterium breve, Clostridium difficile, and Clostridium perfringens were obtained from a collection at the University Hospital of Bordeaux. Strains were identified using matrix-assisted laser desorption ionization–time-of-flight (MALDI–TOF) mass spectrometry. All strains were pre-cultured for 24 h by inoculation on Columbia blood agar (Thermo Scientific, Waltham, MA, USA) under anaerobic conditions (5% H_2_, 10% CO_2_, and 85% N_2_) at 35 °C. Pre-cultures were resuspended in sterile water at a concentration equivalent to the MC Farland 4 opacity standard for each strain. These solutions were mixed at 1:4 dilution with BH broth (Thermo Scientific) containing various concentrations of metformin (0, 5, 10, 20, 50, and 100 mM). Solutions were then inoculated into a 96-well microplate and incubated under anaerobic conditions. The effect of metformin on bacterial growth was analyzed in terms of the optical density at a wavelength of 600 nm (OD_600_) using a Nanodrop microplate reader (BMG Labtech, Champigny-sur-Marne, France) after 24 h of incubation.

## Figures and Tables

**Figure 1 pharmaceuticals-14-00329-f001:**
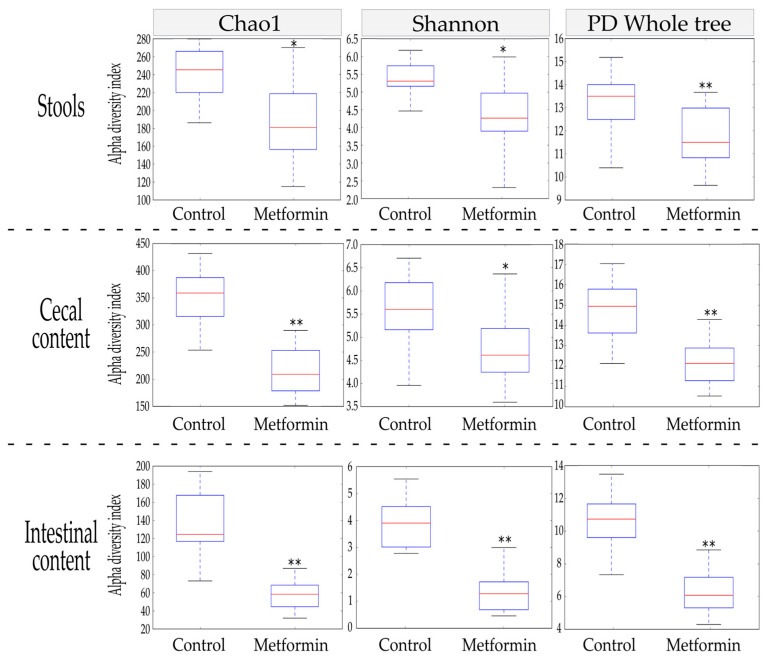
Comparison of alpha diversity indices between metformin (*n* = 20) and control (*n* = 18) treatment groups in stool, cecal, and intestinal content (Student’s *t*-test; * *p* ≤ 0.002; ** *p* ≤ 0.001).

**Figure 2 pharmaceuticals-14-00329-f002:**
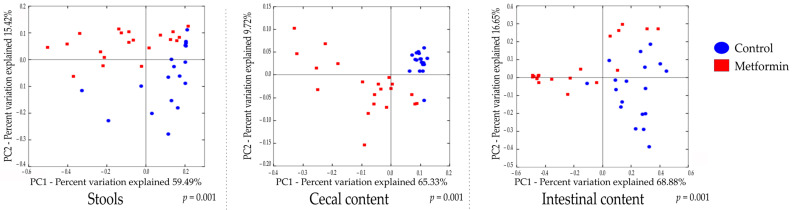
2D principal coordinate analysis (PCoA) plots created using weighted UniFrac distances. Adonis statistical Table 999. permutations; *R*^2^ = 0.224, 0.388, and 0.377 for stool, cecal, and intestinal content, respectively, *p* = 0.001).

**Figure 3 pharmaceuticals-14-00329-f003:**
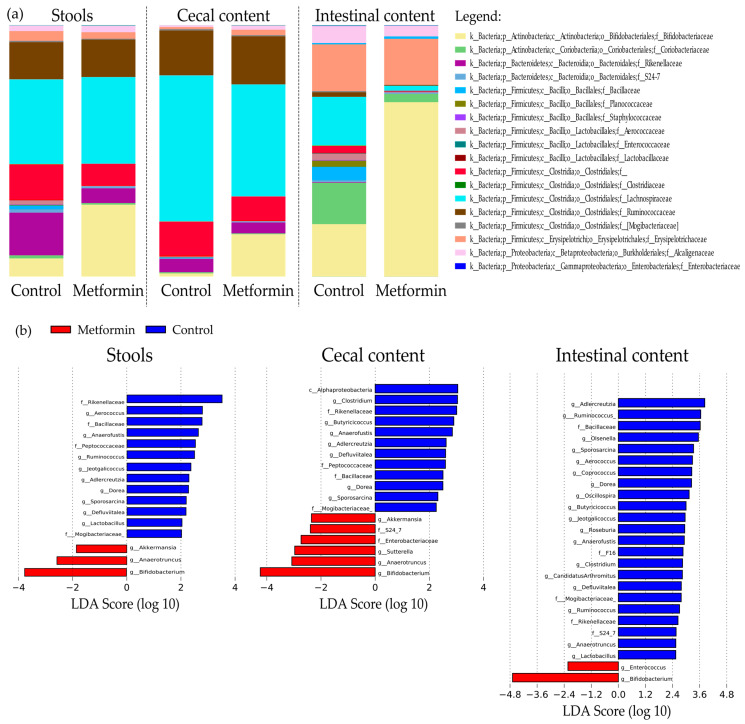
(**a**) Taxonomic distribution of the digestive microbiota in mice infected with *Helicobacter pylori* at the family level. (**b**) Histogram obtained using linear discriminant analysis (LDA) effect size analysis (LEfSe) to identify differentially abundant taxa in fecal, cecal, and intestinal samples. Genera and families with LDA scores >2 were identified as significantly more abundant between the metformin and control groups (*p* < 0.01).

**Figure 4 pharmaceuticals-14-00329-f004:**
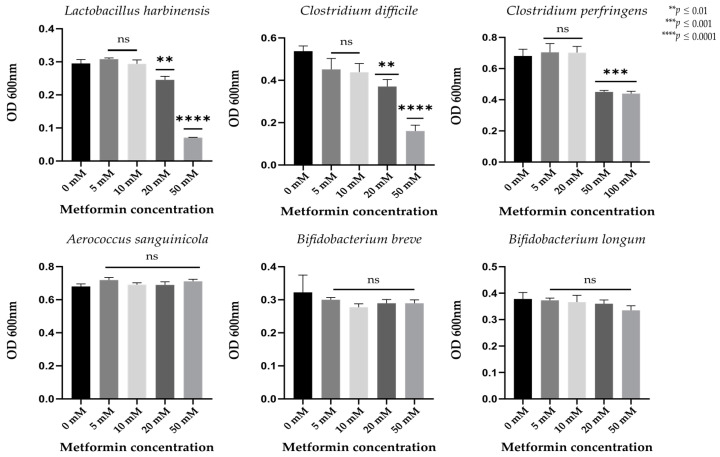
Growth of different bacterial strains at different metformin concentrations after 24 h of anaerobic incubation. Ns, non-significant. ** *p* < 0.01; *** *p* < 0.001; **** *p* < 0.0001 (Student’s *t*-test; *n* = 3).

**Figure 5 pharmaceuticals-14-00329-f005:**
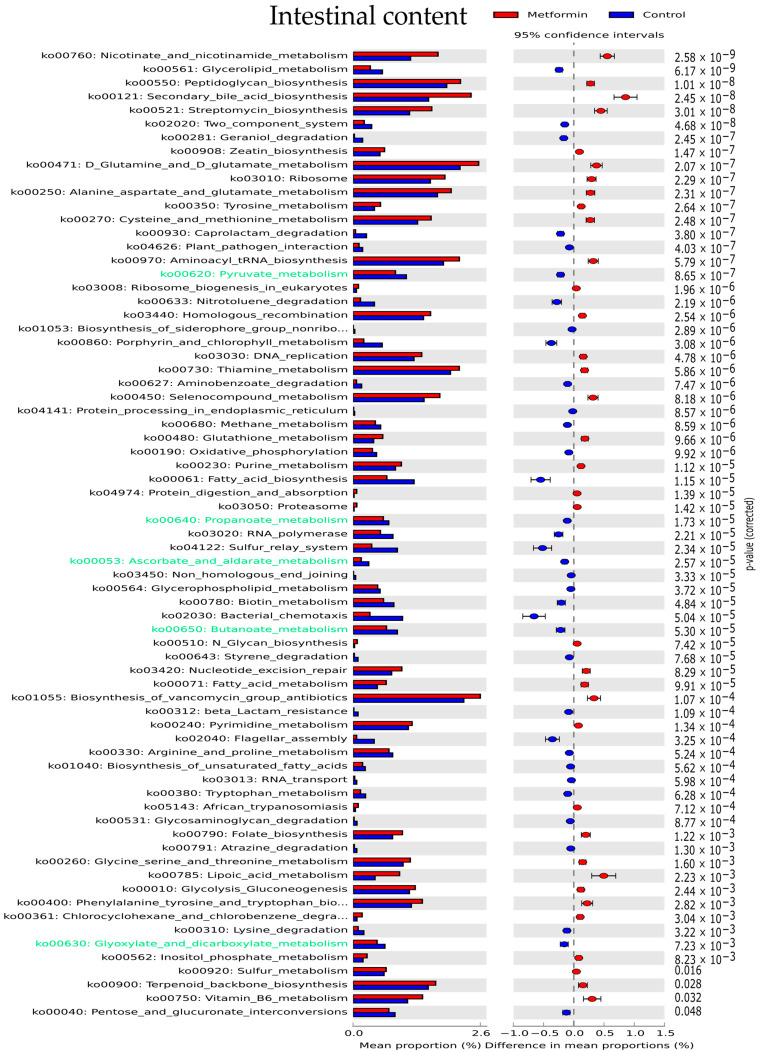
Differentially enriched Kyoto Encyclopedia of Genes and Genomes (KEGG) pathways (relative abundance > 0.001%) in intestinal microbiota (*p* < 0.05). KEGG pathways implicated in carbohydrate metabolism are indicated with green text.

**Figure 6 pharmaceuticals-14-00329-f006:**
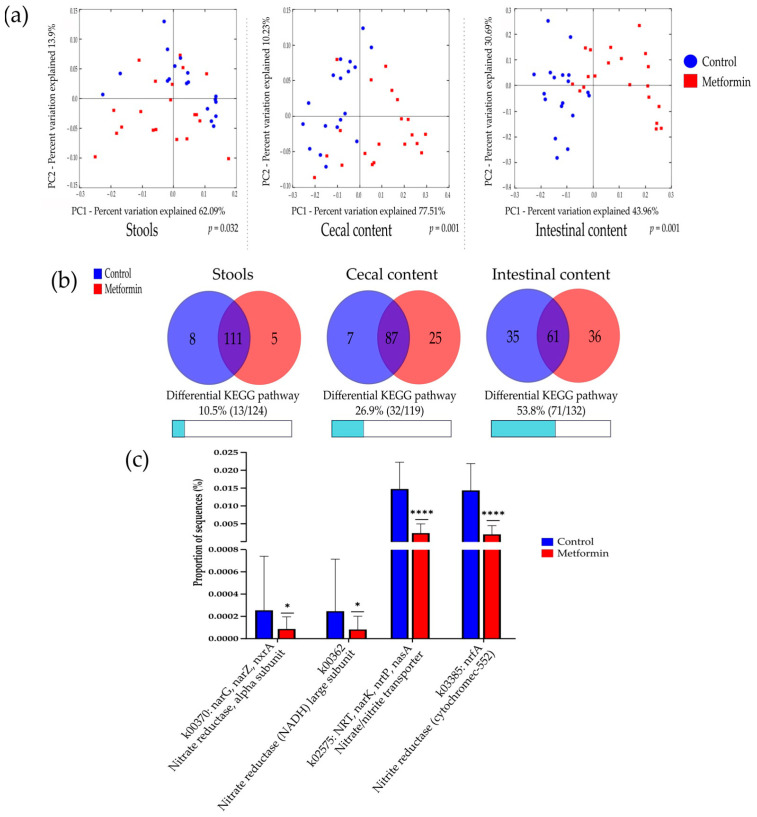
Potential functional pathways in the gut microbiota of metformin and control group mice. (**a**) PCoA plot comparing the microbiota of the metformin and control groups based on potential functional pathways identified using Bray–Curtis distances. Adonis statistical tests showed significant differences between the two groups (999 permutations, *R*^2^ = 0.103, 0.318, and 0.310 for stool, cecal, and intestinal content, respectively, *p* = 0.032 for stool, *p* = 0.001 for cecal and intestinal content). (**b**) Characteristics of differential KEGG pathway numbers in stool, cecal, and intestinal content. Red and blue circles indicate the numbers of KEGG pathways predicted to be relatively overexpressed in the metformin and control group, respectively. (**c**) Proportion of sequences of specific genes coding for nitrate reductase and nitrite reductase enzyme in the intestinal microbiota for the metformin (*n* = 20) and control (*n* = 18) groups. * *p* < 0.05; **** *p* < 0.0001; Mann–Whitney *U* test.

**Table 1 pharmaceuticals-14-00329-t001:** Comparison of the relative abundance of bacteria between the metformin and control treatment groups at all Table 1. 10^−4^ were highlighted, red and blue indicate greater abundance in the metformin and control groups, respectively. Ctrl: control, Met: metformin, NS: non-significant (*p* > 0.05).

	Stools				Caecal Content				Intestinal Content			
		Control (%)	Metformin (%)	*p*		Control (%)	Metformin (%)	*p*		Control (%)	Metformin (%)	*p*
Phylum	*Firmicutes*	71.21	62.02	NS	*Firmicutes*	91.78	74.91	0.000004	*Firmicutes*	54.14	22.63	0.000042
	*Bacteroidetes*	17.94	6.504	0.000068	*Bacteroidetes*	5.637	4.666	NS	*Actinobacteria*	37.08	70.97	0.000015
	*Actinobacteria*	8.519	28.76	0.000564	*Actinobacteria*	1.844	18.53	0.000003	*Proteobacteria*	7.367	5.667	NS
	*Proteobacteria*	2.157	2.455	NS	*Proteobacteria*	0.69	1.682	0.000717	*Bacteroidetes*	1.046	0.4659	0.0296
	*Verrucomicrobia*	0.1462	0.248	NS	*Verrucomicrobia*	0.04292	0.2031	0.000002	*Verrucomicrobia*	0.2619	0.1808	NS
Class	*Clostridia*	64.08	59.39	NS	*Clostridia*	90.33	72.38	<0.000001	*Clostridia*	25.2	2.885	0.000009
	*Bacteroidia*	17.94	6.504	0.000068	*Bacteroidia*	5.637	4.666	NS	*Actinobacteria*	20.79	67.44	<0.000001
	*Actinobacteria*	7.277	28.14	0.000283	*Actinobacteria*	1.393	18.15	0.000002	*Erysipelotrichi*	18.29	19.24	NS
	*Erysipelotrichi*	3.741	2.467	NS	*Erysipelotrichi*	0.8205	2.44	NS	*Coriobacteriia*	16.29	3.529	0.000095
	*Bacilli*	3.386	0.1616	0.00416	*Betaproteobacteria*	0.6795	1.667	0.000663	*Bacilli*	10.65	0.5026	0.018
	*Betaproteobacteria*	2.142	2.427	NS	*Bacilli*	0.6257	0.08574	0.0167	*Betaproteobacteria*	6.716	4.445	NS
	*Coriobacteriia*	1.241	0.6206	0.0301	*Coriobacteriia*	0.451	0.3785	NS	*Bacteroidia*	1.044	0.4628	0.0294
	*Verrucomicrobiae*	0.1462	0.248	NS	*Verrucomicrobiae*	0.04292	0.2031	0.000002	*Alphaproteobacteria*	0.5376	1.076	NS
									*Verrucomicrobiae*	0.2619	0.1808	NS
Order	*Clostridiales*	64.08	59.39	NS	*Clostridiales*	90.33	72.38	<0.000001	*Clostridiales*	25.2	2.885	0.000009
	*Bacteroidales*	17.94	6.504	0.000068	*Bacteroidales*	5.637	4.666	NS	*Bifidobacteriales*	20.79	67.44	<0.000001
	*Bifidobacteriales*	7.277	28.14	0.000283	*Bifidobacteriales*	1.392	18.15	0.000002	*Erysipelotrichales*	18.29	19.24	NS
	*Erysipelotrichales*	3.741	2.467	NS	*Erysipelotrichales*	0.8205	2.44	NS	*Coriobacteriales*	16.29	3.529	0.000095
	*Burkholderiales*	2.142	2.427	NS	*Burkholderiales*	0.6795	1.667	0.000663	*Bacillales*	7.818	0.05888	0.0213
	*Bacillales*	1.779	0.02667	0.0147	*Coriobacteriales*	0.451	0.3785	NS	*Burkholderiales*	6.716	4.445	NS
	*Lactobacillales*	1.606	0.135	0.00112	*Bacillales*	0.3566	0.01136	0.0275	*Lactobacillales*	2.833	0.4437	0.0178
	*Coriobacteriales*	1.241	0.6206	0.0301	*Lactobacillales*	0.2691	0.07438	0.014	*Bacteroidales*	1.044	0.4628	0.0294
	*Verrucomicrobiales*	0.1462	0.248	NS	*Verrucomicrobiales*	0.04292	0.2031	0.000002	*Rickettsiales*	0.5069	1.056	NS
									*Verrucomicrobiales*	0.2619	0.1808	NS
Family	*Lachnospiraceae*	34.13	34.93	NS	*Lachnospiraceae*	57.86	43.3	0.00035	*Bifidobacteriaceae*	20.79	67.44	<0.000001
	*Rikenellaceae*	16.58	5.729	0.000128	*Ruminococcaceae*	18.02	18.98	NS	*Lachnospiraceae*	19.8	2.11	0.000009
	*Ruminococcaceae*	14.88	15.05	NS	*Rikenellaceae*	5.371	4.218	NS	*Erysipelotrichaceae*	18.29	19.24	NS
	*Bifidobacteriaceae*	7.277	28.14	0.000283	*Bifidobacteriaceae*	1.392	18.15	0.000002	*Coriobacteriaceae*	16.29	3.529	0.000095
	*Erysipelotrichaceae*	3.741	2.467	NS	*Erysipelotrichaceae*	0.8205	2.44	NS	*Alcaligenaceae*	6.716	4.445	NS
	*Alcaligenaceae*	2.138	2.427	NS	*Alcaligenaceae*	0.6795	1.667	0.000663	*Bacillaceae*	5.125	0.002976	0.0411
	*Aerococcaceae*	1.521	0.1177	0.00143	*[Mogibacteriaceae]*	0.579	0.473	0.0326	*Aerococcaceae*	2.499	0.2731	0.0228
	*Bacillaceae*	1.361	0	0.019	*Coriobacteriaceae*	0.451	0.3785	NS	*Planococcaceae*	2.439	0.001683	0.00703
	*S24-7*	1.358	0.7749	0.0403	*S24-7*	0.266	0.4476	0.0237	*Ruminococcaceae*	1.95	0.3866	0.00399
	*Coriobacteriaceae*	1.241	0.6206	0.0301	*Bacillaceae*	0.2608	0	0.0373	*S24-7*	0.7812	0.4473	NS
	*[Mogibacteriaceae]*	0.548	0.3557	0.00734	*Aerococcaceae*	0.2524	0.0623	0.013	*[Mogibacteriaceae]*	0.4774	0.03272	0.0223
	*Planococcaceae*	0.2407	0	0.0147	*Verrucomicrobiaceae*	0.04292	0.2031	0.000002	*Rikenellaceae*	0.2633	0.01555	0.0232
	*Verrucomicrobiaceae*	0.1462	0.248	NS					*Verrucomicrobiaceae*	0.2619	0.1808	NS
									*Staphylococcaceae*	0.2544	0.05422	NS
									*Lactobacillaceae*	0.2505	0.0467	0.00019
									*Enterococcaceae*	0.08181	0.1239	NS

## Data Availability

Data is contained within the article or Appendix A.

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
