# Peer review of "Metformin Modifies the Gut Microbiota of Mice Infected with Helicobacter pylori"

_pharmaceuticals, 2021, doi:10.3390/ph14040329_

Round 1

Reviewer 1 Report

I appreciate the work and efforts of Jauvain and colleagues regarding my comments and suggestions to improve their manuscript. However, I still have some questions and comments.

QIIME 1.9.1 is for some applications such as diagnostics with fixed protocol consistency in analysis tools and strategy fundamental. Especially when results should be comparable and consistent over time. But for the analysis of 16S data to investigate/answer scientific questions I strongly recommend the usage of QIIME2 and especially an up to date reference database. Bailén et al. (2020) used in their pipeline comparison for every pipeline the same Greengenes database. Which is in my opinion the main reason for the similar taxonomic results between the different pipelines and approaches.

Just to be clear here, at the end of rarefication all the samples comprise the same number of sequences. From showing a rarefaction plot no one can derive the common min. number of sequences for each sample (rarefication depth). Fig. S4 shows some samples with about 30000 seqs per sample and others with about 50000. Readers can see, that sequencing depth is sufficient for all the samples as seq depth presented in S4 is greater than 30000 seq for all samples. Please include somewhere in the manuscript to which number of sequences the samples have been rarefied to, if data was rarified (which is usually the case for alpha- and beta-diversity calculations).

There seems to be a discrepancy between Fig. S4 and Table S3 or Table S2 - Table Number is ambiguous. In Fig S4. max num of seqs per sample is about 50000. BUT according to Table S3 or S2 eg Sample 1 comprises 116160 after pre-processing. Almost all the samples of the presented table comprise more than 50000 samples. Please clarify.

Authors should at least integrate a project accession of ENA. This will take less than 5 minutes to create this page. Sequences can be added or released upon the acceptance of the manuscript.

P16 L306-307 Authors should clarify entire intestinal content (without cecum) also in the manuscript to avoid ambiguity. Currently this information is only an answer to one of my questions.

Fig 3b. should be improved rearranged. Headline intestinal content is aligned above (b). All axes should be aligned horizontally.

Fig 5a. Strongly recommend here also 2D instead of 3D plots. In addition, proportion of sequences which are added to functional pathways is very very low. So, entropy of this significant differences is questionable.

Author Response

I appreciate the work and efforts of Jauvain and colleagues regarding my comments and suggestions to improve their manuscript. However, I still have some questions and comments.

QIIME 1.9.1 is for some applications such as diagnostics with fixed protocol consistency in analysis tools and strategy fundamental. Especially when results should be comparable and consistent over time. But for the analysis of 16S data to investigate/answer scientific questions I strongly recommend the usage of QIIME2 and especially an up to date reference database. Bailén et al. (2020) used in their pipeline comparison for every pipeline the same Greengenes database. Which is in my opinion the main reason for the similar taxonomic results between the different pipelines and approaches.

We managed to obtain new data using the SILVA database and we started to use QIIME2. For the moment, we just had time to obtain alpha-diversity indices (chao1 and Shannon). Similar results were obtained as those obtained with the Greengenes database and QIIME1.9.1. If mandatory for publication, we will re-analyse the whole data with QIIME2 but we need more times to review the entire article.

Just to be clear here, at the end of rarefication all the samples comprise the same number of sequences. From showing a rarefaction plot no one can derive the common min. number of sequences for each sample (rarefication depth). Fig. S4 shows some samples with about 30000 seqs per sample and others with about 50000. Readers can see, that sequencing depth is sufficient for all the samples as seq depth presented in S4 is greater than 30000 seq for all samples. Please include somewhere in the manuscript to which number of sequences the samples have been rarefied to, if data was rarified (which is usually the case for alpha- and beta-diversity calculations).

Thank you for this precision, you’re right the samples have been rarefied to 32606 sequences. Following your recommendation, this information has been added in the manuscript (Page 15, Line 334).

There seems to be a discrepancy between Fig. S4 and Table S3 or Table S2 - Table Number is ambiguous. In Fig S4. max num of seqs per sample is about 50000. BUT according to Table S3 or S2 eg Sample 1 comprises 116160 after pre-processing. Almost all the samples of the presented table comprise more than 50000 samples. Please clarify.

We apologize for this mistake; the incorrect number has been modified.  
The number of sequences initially showed in Table S3 was the number of sequences after demultiplexing step. We modified the number in the table (now Table S4) with the number of sequences after all the pre-processing steps. After this modification, the smallest number of sequences was that of sample n° 97 with 32606 sequences, on average number of sequences per sample was 67 502. Figure S4 represented rarefaction curves showing the number of observed OTUs as a function of the number of sequences per samples, the maximum number of sequences per sample was showed until 50 000 because rarefaction curve level out from 20 000 seqs per sample and it was sufficient to see if our diversity was correctly sampled.  

Authors should at least integrate a project accession of ENA. This will take less than 5 minutes to create this page. Sequences can be added or released upon the acceptance of the manuscript.

Thanks to your recommendation the 16S rRNA sequencing datasets generated in this study can be found in the SRA database (http://www.ncbi.nlm.nih.gov/bioproject/701274). This information has been added Page 15– Line 344. 

P16 L306-307 Authors should clarify entire intestinal content (without cecum) also in the manuscript to avoid ambiguity. Currently this information is only an answer to one of my questions.

We understand your recommendation and this information has been clarified in the manuscript (Page 14 Line 306).

Fig 3b. should be improved rearranged. Headline intestinal content is aligned above (b). All axes should be aligned horizontally.

Thank you for this suggestion, Fig 3b has been improved.

Fig 5a. Strongly recommend here also 2D instead of 3D plots. In addition, proportion of sequences which are added to functional pathways is very very low. So, entropy of this significant differences is questionable.

We followed your suggestion and modified 3D in 2D plots in Figure 5a.
Concerning the functional pathways and the proportion of sequences, we understand your remark but genes coding for nitrate reductase seems to be little represented in microbiota, however our results concerning proportion of sequences were similar to that showed by Ferreira et al., 2018 in Gut. In our opinion, it was an interesting information to be mentioned and as we specified in the manuscript, more experiments should be performed to understand the metabolic modifications induced with metformin microbiota changes.  

Reviewer 2 Report

Major revisions:

1) Page 3, lines 66-67. It is known that a high diversity and richness of intestinal microbiota are considered markers of healthy microbiota, as the Authors themselves report in the manuscript (lines 221-222). Please the Authors discuss their data, reporting these parameters significantly reduced.

2) By following the previous suggestion, the Authors report, line 306, that They collected stool samples before H. pylori infection. No data are reported on these samples, which I consider very important. I believe that, to better discuss the data obtained after metformin treatment, data on biological samples from H. pylori-untreated mice should be considered as the primary "Control". Please the Authors discuss it.

3) The Authors adopted an experimental animal model treating the mice with metformin, a drug crucial for glucose metabolism. Unfortunately, the Authors do not report any data on this parameter. Please the Authors discuss it.

4) Why did the Authors choose female, five week old mice? Wouldn't it be easier to use male mice to study the effects of metformin on gut microbiota composition and on metabolic pathways? Indeed, considering the age of the mice and the treatment time (3 days for H. pylori and 12 days for metformin treatment), the mice reach sexual maturity at 7 week old, that could interfere with metformin-induced metabolic pathways and gut microbiota composition. Please discuss it.

Minor revisions:

1) Page 2, lines 37-38 and lines 39-40: the Authors refer to reference 5 as published in 2005, instead it is from 2015, while reference 6, referred to as “recent studies”, has been published on 2005. Exchange of references with little attention or the Authors have a scientific purpose?. Please the Authors clarify it.

2) Line 53. The Authors speak of “decrease in gastric mucosa diversity”, reporting the reference 17. Did the Authors mean bacterial diversity?

3) Referring to line 66 and comparing the data in this paragraph, in particular Tab. 1, it does not appear that the data always indicate a reduction in bacterial percentages at all levels (Phylum, Class, Order and Family), in all three samples studied (stools, cecal content and intestinal content). Would a more careful description of the Table be necessary? The Authors discuss it.

4) Page 11, line 218. As previously reported, the Authors report a decreased richness and diversity of the gut microbiota in metformin-treated mice. A more careful interpretation of the data, considering my observations of point 3 and the suggestions of point 2 of the Major revisions perhaps should be introduced.

5) Line 235, page 12: the Authors discuss their data emphasizing the inhibition of Lactobacillus and Clostridium. Considering the different taxa of these bacteria, some of which are defined as "good bacteria", which are those inhibited? Please the Authors discuss their data in view of the side effects on gut microbiota following metformin treatment in human patients.

6) Page 14, line 301, the Authors report having treated mice with a highly concentrated suspension of H. pylori. Please explain what means by highly.

Author Response

Major revisions:

1) Page 3, lines 66-67. It is known that a high diversity and richness of intestinal microbiota are considered markers of healthy microbiota, as the Authors themselves report in the manuscript (lines 221-222). Please the Authors discuss their data, reporting these parameters significantly reduced.

Thank you for this interesting remark. As you correctly mentioned, the present study showed that metformin decreased the richness and diversity of mice gut microbiota. We were not very surprised to observe that metformin decreased the richness and diversity of microbiota. Indeed, a previous work performed in our lab demonstrated that metformin inhibits directly the growth of H. pylori (Courtois et al., 2018) and other studies performed have already demonstrated the metformin direct antimicrobial effect on several gut bacteria (Maier et al., 2018). High microbiota diversity and richness are usually considered to be markers of a healthy microbiota, but even if the richness and the diversity of the gut microbiota decreased in our study, metformin treatment led to the increase of Akkermansia muciniphila which is associated with metabolic improvement. Furthermore, metformin treatment led to an increase of Bifidobacterium in our study and several studies reported beneficial effects from this normal component of gastro-intestinal tract. It is possible that the decreases in the abundance of any bacterial taxon may lead to the relative emergence of metabolically beneficial microorganisms. We modified sentences order and added precisions in Discussion parts for more clarity (Page 11– Line 232 and Page 12 – Line 265).

2) By following the previous suggestion, the Authors report, line 306, that They collected stool samples before H. pylori infection. No data are reported on these samples, which I consider very important. I believe that, to better discuss the data obtained after metformin treatment, data on biological samples from H. pylori-untreated mice should be considered as the primary "Control". Please the Authors discuss it.

Thank you for this question. We collected stools before H. pylori infection and obviously before metformin treatment. No differences concerning alpha-diversity and beta-diversity were observed between the different mice groups: these data are presented in Figure S1 (Supplementary Data), confirming that groups were comparable. We also collected stools just before the beginning of treatment, 3 days after H. pylori infection but we didn’t report these data because here again no difference was observed in alpha and beta diversity (see the following Figure). À voir si ajout dans le texte en mettant data not shown

Figure: Alpha and beta diversity comparison of fecal microbiota between the metformin and control groups after infection with H. pylori and before the beginning of treatment. (a) Alpha diversity analyses. Ns, non-significant. Student’s t-test. (b) Principal coordinate analysis (PCoA) plots created using weighted UniFrac distances. Green and yellow dots indicate metformin and control samples, respectively.

3) The Authors adopted an experimental animal model treating the mice with metformin, a drug crucial for glucose metabolism. Unfortunately, the Authors do not report any data on this parameter. Please the Authors discuss it.

Thank you for this comment. You are right, metformin has important impact on glucose metabolism. But, here we didn’t report any data concerning glucose metabolism because the aim of our study was to describe metformin effects on microbiota of mice infected with H. pylori. Indeed, several studies have already analyzed glucose metabolism in mice treated with metformin and they reported a decrease of body weight and total cholesterol level. In mice with high fat diet, metformin decreases serum glucose level (Lee and Ko, 2014), (Martin-Montalvo et al., 2013). These studies are particularly interesting to better understand metformin effect on diabetes.  In our experiment, we measured the weight of our mice after treatments, no significant differences were observed in weight between mice treated with metformin and mice treated with PBS. Here, we chose to work with healthy mice in term of glucose metabolism, with a normal body weight and a normal diet and we focused on metformin effects on gut microbiota in digestive cancer prevention. However, we are completely aware that microbiota modification observed with metformin treatment are partly the effect of metformin on glucose metabolism. As we mentioned Page 12 - Line 238; metformin treatment was revealed to enhance the release of glucose into the intraluminal space of the intestine in humans; therefore, high glucose concentration in the intestinal lumen may impact bacterial development. In this study, we suggested that metformin-induced microbiota changes are probably the result of both direct and physiological effects of metformin.

4) Why did the Authors choose female, five week old mice? Wouldn't it be easier to use male mice to study the effects of metformin on gut microbiota composition and on metabolic pathways? Indeed, considering the age of the mice and the treatment time (3 days for H. pylori and 12 days for metformin treatment), the mice reach sexual maturity at 7 week old, that could interfere with metformin-induced metabolic pathways and gut microbiota composition. Please discuss it.

We chose female mice for this study because cages were constituted of 5 mice and, in our experience, female live more easily in groups. Males could be more aggressive with their congeners, we wanted to limit conflicts and risk of injuries. Concerning the age of mice, you are right, metformin treatment was administered during 12 days and the treatment period was a little bit more than 2 weeks. At the end of the treatment, mice were aged of 8 weeks and 2 days. We understand your remark but these mice were initially used for another experiment which did not implicate microbiota study. Following the initial experimental results observed, we decided to perform an analysis of the digestive microbiota on the samples collected from these mice. However, mice were all five-week-old at the beginning and the same hormonal modifications were operated in the 2 mice groups. Thus, differences observed are due to the treatment.

Minor revisions:

1) Page 2, lines 37-38 and lines 39-40: the Authors refer to reference 5 as published in 2005, instead it is from 2015, while reference 6, referred to as “recent studies”, has been published on 2005. Exchange of references with little attention or the Authors have a scientific purpose?. Please the Authors clarify it.

Thank you for this interesting remark. On line 37-38 we mentioned: “Since 2005, metformin has also been studied intensively for its anti-tumor properties in different types of cancer [5].”
The study mentioned as reference 5 is a review published in 2015 by Morales et al. about the potential for metformin in oncology. This article lists the studies performed since 2005, which were effectively the year where Evans et al. (reference 6) published the first observational study reporting metformin for cancer prevention.     
Concerning the next sentence: “Recent studies have hypothesized that metformin treatment may reduce cancer risk in diabetic patients [6].”
Here, reference 6 really refers to the article of Evans et al. published in 2005, we had chosen to refer to the first study that hypothesized that metformin treatment may reduce cancer risk in diabetic patients. But you’re completely right, “recent studies” were not adapted.
To be more readable, we modified these sentence in the manuscript (Page 2, Line 37-40).

2) Line 53. The Authors speak of “decrease in gastric mucosa diversity”, reporting the reference 17. Did the Authors mean bacterial diversity?

Thank you for this remark, we modified this sentence (Page 2 – Line 54) for more clarity.

3) Referring to line 66 and comparing the data in this paragraph, in particular Tab. 1, it does not appear that the data always indicate a reduction in bacterial percentages at all levels (Phylum, Class, Order and Family), in all three samples studied (stools, cecal content and intestinal content). Would a more careful description of the Table be necessary? The Authors discuss it.

Thank you for this interesting remark, you are right, metformin treatment decreased some bacterial taxa abundance but this decrease was for the benefit of other bacterial taxa which abundance increased. We added more precisions in the manuscript about this point (Page 5 – Line 122 and 127 and Page – Line 134).

4) Page 11, line 218. As previously reported, the Authors report a decreased richness and diversity of the gut microbiota in metformin-treated mice. A more careful interpretation of the data, considering my observations of point 3 and the suggestions of point 2 of the Major revisions perhaps should be introduced.

Thank you for this remark, as we answered to your suggestion of point 1 of the Major revision, in our opinion the decrease of any bacterial taxa abundance (linked to a decrease of richness and diversity) induced by metformin treatment is not necessarily associated with an unhealthy microbiota. Indeed, decreases in the abundance of any bacterial taxa may lead to the relative emergence of metabolically beneficial microorganisms (Akkermansia muciniphila or Bifidobacterium for example in this study). This notion has been added more clearly in the manuscript (Page 11– Line 232 and Page 12 -Line 265)

5) Line 235, page 12: the Authors discuss their data emphasizing the inhibition of Lactobacillus and Clostridium. Considering the different taxa of these bacteria, some of which are defined as "good bacteria", which are those inhibited? Please the Authors discuss their data in view of the side effects on gut microbiota following metformin treatment in human patients.

You are right, in this study we showed that metformin directly inhibited the growth of Lactobacillus and Clostridium gut bacteria, which could be defined, especially for Lactobacillus as “good bacteria”. Effect of metformin on gut microbiota is very complex because some “good bacteria” could be inhibited by treatment, but, as we mentioned before, other good bacteria are increased in response to metformin treatment (Akkermansia muciniphila or Bifidobacterium). We don’t have any data with this experiment, but as you suggest, we can absolutely hypothesize that metformin digestive side effects in human patients are the result of gut modification induced by the drug, included the decrease of “good bacteria” in response to the treatment

6) Page 14, line 301, the Authors report having treated mice with a highly concentrated suspension of H. pylori. Please explain what means by highly.

The highly concentrated suspension of H. pylori corresponded to McFarland 7 opacity standard as used in previous studies using H. pylori infection mice model. We added this information in the manuscript (Page 14 – Line 302). 

Reviewer 3 Report

In the manuscript of Jauvain et al. the metformin effect on gut microbiota of mice infected with H. pilori was investigated. Although the investigation of gastric microbiota would complete the work, the research described in the present manuscript is innovative and well design. However, in my opinion, some improvements are needed and suggested below.

Line 18 and 61: ‘three different digestive sites’ please indicate which ones.

Line 24, 93, 216, 253, 295, and 296: for clarity, please change ‘digestive’ with ‘gut’, since here the gastric microbiota was not analyzed.

Line 68: ‘PD’ please describe the acronyms the first time you cite them in the text.

Line 77: please explain why the two groups are not numerically uniform, already from the randomization.

Line 142-143 and 227: please associate some data to this sentence, the species were never shown in the data.

Line 294: please add the limitations of the study, such as the analyzed gastric microbiota.

Line 300: the authors should justify why they used only female mice and indicate this as a limitation of the study.

Line 302: divided how? Randomly? Please add.

Line 304: “10 mg/mouse” the authors should explain how they chose this amount.

Line 319: “using Genoscreen” Genoscreen is a company, so maybe ‘performed by’ is more suitable.

Suppl. Mat. S1 and line 330: “Qiime 1.9.1 pipeline”. Qiime 2 has been released for three years, so why the authors still use Qiime 1?

Suppl. Mat.: several English mistakes are present in this file, please correct.

Author Response

In the manuscript of Jauvain et al. the metformin effect on gut microbiota of mice infected with H. pilori was investigated. Although the investigation of gastric microbiota would complete the work, the research described in the present manuscript is innovative and well design. However, in my opinion, some improvements are needed and suggested below.

Line 18 and 61: ‘three different digestive sites’ please indicate which ones.

Thank you for this suggestion, we added this information in the manuscript (Page 1- Line 18 and Page 2- Line 62).

Line 24, 93, 216, 253, 295, and 296: for clarity, please change ‘digestive’ with ‘gut’, since here the gastric microbiota was not analyzed.

You’re right, we made these modifications for clarity (Page 1 – Line 24, Page 4 – Line 95, Page 11 – Line 219, Page 13 – Line 256, Page 14– Line 298).

Je n’ai pas modifié la dernière (296), c’est la dernière phrase de la discussion : « These modifications could be implicated in digestive cancer prevention. » je ne pense pas que ce soit faux de laisser « digestive » puisque dans le papier de Sarah il est montré que la metformine diminue la colonisation par HP donc aussi un effet potentiel sur la prévention du cancer gastrique, qu’en penses-tu ?  oui mais s’il le demande on peut le laisser…

Line 68: ‘PD’ please describe the acronyms the first time you cite them in the text.

Thank you for this remark, PD is for phylogenetic diversity, we added the description of this acronym (Page 3 – Line 70).

Line 77: please explain why the two groups are not numerically uniform, already from the randomization.

You’re right, there were initially 20 mice in either metformin and control group, but 2 mice from control group died before the beginning of treatment. Therefore, 18 mice constituted control group and 20 mice constituted metformin group. This information has been clarified in Materials and Methods parts (Page 14 - Line 306).

Line 142-143 and 227: please associate some data to this sentence, the species were never shown in the data.

Thank you for this remark. We have decided to show mainly data at family level because it was the lowest level with a sufficient percentage of identification.

In this study, on average 91% of bacterial taxa identified could be categorized at family level compared with 54% at genus level. However, for an average of 28% of bacterial taxa, species identification was possible. It was the case for the species Akkermansia muciniphila and Bifidobacterium pseudolongum. Following your suggestion, we added the data about species in Table S1.

Line 294: please add the limitations of the study, such as the analyzed gastric microbiota.

Thank you for this suggestion. We added the limitations of the study in the manuscript (Page 14 – Line 297).

Line 300: the authors should justify why they used only female mice and indicate this as a limitation of the study.

We chose female mice for this study because cages were constituted of 5 mice and, in our experience, female lives more easily in groups. Males could be more aggressive with their congeners, we wanted to limit conflicts and risk of injuries. As you suggest, this information was added in Materials and Methods part (Page 14 – Line 306) and also in the limitations of this study (Page 14 – Line 299).

We do not consider that use of female is a limitation since we used homogenous mice to determine impact of metformin on gut microbiota.

Line 302: divided how? Randomly? Please add.

You’re right, groups were divided randomly, we added this information in Material and Methods part (Page 14 – Line 309).

Line 304: “10 mg/mouse” the authors should explain how they chose this amount.

Thank you for this remark. This dosage was determined using the method of dose conversion between human and animal studies. With this method, 10 mg/mouse/day corresponds to 2.4 g of metformin/day for a human adult. The maximal dosage of metformin used to treat Type 2 diabetes patients is 3 g/day. We added this information in Material and Methods part (Page 14 – line 319)

Line 319: “using Genoscreen” Genoscreen is a company, so maybe ‘performed by’ is more suitable.

You’re right, “performed by” is more suitable, we modified this sentence in the manuscript (Page 14 – Line 328).

Suppl. Mat. S1 and line 330: “Qiime 1.9.1 pipeline”. Qiime 2 has been released for three years, so why the authors still use Qiime 1?

Concerning preprocessing, clustering and classification: these steps were performed by Genoscreen, an external private company of our lab. We contacted the engineer about this choice. They used the version 1.9.1 because the Metabiote 2.0 protocol was fixed for this specific version. Furthermore, Genoscreen had designed this process following quality standard requirements and their methodology in this version is qualified for "Good Clinical Laboratory Practice”. It is not possible for Genoscreen to perform the preprocessing, clustering and classification with this new version of QIIME 1.9.1. And this is why unfortunately we were not able to re-analyse the data.

Anyway, thanks to this comment, we checked that QIIME 1.9.1 version was a good pipeline for Illumina platform analysis. Bailén et al. (2020) analyse a known bacterial community (mock) in a pilot study using four different analysis pipelines included QIIME1.9.1 and QIIME2 and compared results obtained.

The study concludes that QIIME 1.9.1 showed quality results and is a good pipeline for Illumina platforms results. No statistical differences were found between observed and expected taxa from the mock up to genus level among the different pipeline tested.

They reported that differences can be observed between the 2 pipelines but they didn’t conclude that QIIME 1.9.1 is obsolete. Furthermore, lots of publication were performed using QIIME 1.9.1 and allowed to discover very interesting informations (à laisser ?)  

Suppl. Mat.: several English mistakes are present in this file, please correct.

Thank you, we made the corrections.

Round 2

Reviewer 1 Report

Thx to the authors for addressing all of my comments and concerns. There is still a minor issue with the SRA link at line 347. There is a type within the provided SRA link, please adjust. (Hint: https is spelled incorrect.)  

Author Response

Thx to the authors for addressing all of my comments and concerns. There is still a minor issue with the SRA link at line 347. There is a type within the provided SRA link, please adjust. (Hint: https is spelled incorrect.)

Thank you for this remark, the correction has been made (Page 15 -Line 351).  

Reviewer 2 Report

Thank you for your answers to my considerations, unfortunately I do not consider some of your answers to be sufficient and scientifically acceptable (n ° 4). The study could be interesting but, in my opinion, the data obtained without any reference to the contribution of sex hormones and the length of the study cannot be considered sufficiently evaluated. There are no considerations on the lack of effects on mouse weight or carbohydrate metabolism. Other studies on healthy mice report different variations of the intestinal microbiota.

Author Response

Review 2:

Thank you for your answers to my considerations, unfortunately I do not consider some of your answers to be sufficient and scientifically acceptable (n ° 4). The study could be interesting but, in my opinion, the data obtained without any reference to the contribution of sex hormones and the length of the study cannot be considered sufficiently evaluated. There are no considerations on the lack of effects on mouse weight or carbohydrate metabolism. Other studies on healthy mice report different variations of the intestinal microbiota.

We thank the reviewer for its interesting comments. Metformin is indeed a drug crucial for glucose metabolism and its use in human can lower body weight. This parameter was not considered in the present study which only focused on microbiota changes observed after a short period of treatment. We did not observe any differences in term of weight after 2 weeks of treatment but it is certainly due to the short period of treatment. We added these precisions into the manuscript (Page 14 – Line 332). Concerning carbohydrate metabolism, as we only focused on microbiota, we didn’t collect any blood during the experiment and consequently we cannot study modifications occurring in this pathway.

In order to better discuss our results by considering sex hormones and sexual maturity of our female mice, references and sentences have been added into the discussion as follows (Page 14 – Line 308):

“Limitations of this study were the lack of information about gastric microbiota; more experiments should be performed to understand the metabolic modifications induced with metformin microbiota changes. We used female mice, which are less aggressive than male and easier to use in animal facilities. Consequently, results obtained are only valid in female and cannot completely be extrapolated to male since there are few differences in the composition of gut microbiota between genders and between female of different hormonal status [46,47]. However, female from either control or metformin group had the same age at the beginning and during all the length of the experiment; therefore, mice from the two groups were exposed to the same sexual hormonal modifications, allowing the groups comparison.”

Another study performed by Ma et al. (2018) on healthy mice, treated with either metformin or saline solution, demonstrated some similar results (specifically concerning the effects on Lachnospiraceae and Verrumicrobiaceae). Concerning the differences observed between our experiment and this other study, it may be due to different experimental parameters like the food or, as you interestingly mentioned, the sex or the age of mice selected.

Reviewer 3 Report

The authors addressed my comments.

Author Response

We thank the reviewer.

Round 3

Reviewer 2 Report

Dear Authors,

I apologize for the firmness of my decisions, thank you for your answers, I continue to be of the opinion that your manuscript, as presented, is not scientifically sufficient to be published in Pharmaceuticals journal.